

# Nine time steps: ultra-fast statistical consistency testing of the Community Earth System Model (pyCECT v3.0)

Daniel J. Milroy[1], Allison H. Baker[2], Dorit M. Hammerling[2], and Elizabeth R. Jessup[1]

[1]University of Colorado, Boulder, CO, USA
[2]The National Center for Atmospheric Research, Boulder, CO, USA

*Correspondence to:* Daniel Milroy (daniel.milroy@colorado.edu)

**Abstract.**

The Community Earth System Model Ensemble Consistency Test (CESM-ECT) suite was developed as an alternative to requiring bitwise identical output for quality assurance. This objective test provides a statistical measurement of consistency between an accepted ensemble created by small initial temperature perturbations and a test set of CESM simulations. In this work, we extend the CESM-ECT suite by the addition of an inexpensive and robust test for ensemble consistency that is applied to Community Atmospheric Model (CAM) output after only nine model time steps. We demonstrate that adequate ensemble variability is achieved with instantaneous variable values at the ninth step, despite rapid perturbation growth and heterogeneous variable spread. We refer to this new test as the Ultra-Fast CAM Ensemble Consistency Test (UF-CAM-ECT) and demonstrate its effectiveness in practice, including its ability to detect small-scale events and its applicability to the Community Land Model (CLM). The new ultra-fast test facilitates CESM development, porting, and optimization efforts, particularly when used to complement information from the original CESM-ECT suite of tools.

**Keywords.** Community Earth System Model, Ensemble Consistency Test, statistical consistency, Community Atmosphere Model, Community Land Model, non-bit-for-bit, initial time step, ensemble spread

## 1 Introduction

Requiring bit-for-bit (BFB) identical output for quality assurance of climate codes is restrictive. The codes are complex and constantly evolving, necessitating an objective method for assuring quality without BFB equivalence. Once the BFB requirement is relaxed, evaluating the possible ways datasets can be distinct while still representing similar states is nontrivial. Baker et al. (2015) addresses this challenge by considering statistical distinguishability from an ensemble for the Community Earth System Model (CESM) (Hurrell et al., 2013), an open source Earth System Model (ESM) developed principally at the National Center for Atmospheric Research (NCAR). Baker et al. (2015) developed the CESM ensemble consistency test (CESM-ECT) to address the need for a simple and objective tool to determine whether non-BFB CESM outputs are statistically consistent with the expected output. Substituting statistical indistinguishability for BFB equivalence allows for more aggressive code optimizations, more efficient algorithms, and heterogeneous execution environments.





CESM-ECT is a suite of tools which measures statistical consistency by focusing on 12 month output from two different component models within CESM: the Community Atmospheric Model (CAM), and the Parallel Ocean Program (POP), with ensemble consistency testing tools referred to as CAM-ECT and POP-ECT, respectively. The key idea of CESM-ECT is to compare new non-BFB CESM output (e.g., from a newly built machine) to an ensemble of simulation outputs from the original

or "accepted" configuration (e.g., a trusted machine and software and hardware configuration), quantifying their difference by an objective statistical metric. CESM-ECT issues a pass for the newly generated output if it is statistically indistinguishable from the ensemble's distribution, and a fail if the results are distinct. The selection of a representative or "accepted" ensemble is critical to CESM-ECT's determination of whether new simulations pass. The question of the ensemble composition and size for CAM-ECT is addressed in Milroy et al. (2016), which concludes that ensembles created by aggregating sources of

variability from different compilers improve the classification power and accuracy of the test. At this time, CESM-ECT is used by CESM software engineers and scientists for both port verification and quality assurance for code modification and updates. In light of the success of CAM-ECT, the question arose as to whether the test could also be performed using a time period shorter than one year, and in particular, just a small number of time steps.

The effects of rounding, truncation and initial condition perturbation on chaotic dynamical systems is a well studied area of

research with foundations in climate science. The growth of initial condition perturbations on CAM has been investigated since Rosinski and Williamson (1997), whose work resulted in the PerGro test. This test examined the rate of divergence of CAM variables at initial time steps between simulations with different initial conditions. The rates were used to compare the behavior of CAM under modification to that of an established version of the model in the context of the growth of machine roundoff error. With the advent of CAM5, PerGro became less useful for classifying model behavior, as the new parameterizations in

the model resulted in much more rapid spread. Accordingly, it was commonly held that using a small number of time steps was an untenable strategy due to the belief that the model's initial variability (that far exceeded machine roundoff) was too great to measure statistical difference effectively. Indeed, even the prospect of using runs of one simulation year for CESM-ECT was met with initial skepticism. Note that prior to the CESM-ECT approach, CESM verification was a subjective process predicated on climate scientists' expertise in analyzing multi-century simulation output. The success of CESM-ECT's technique of using

properties of yearly CAM means (Baker et al., 2015) translated to significant cost savings for verifying the model.

Motivated by the success and cost improvement of CESM-ECT we were curious as to whether its general technique could be applied after a few initial time steps, in analogy with Rosinski and Williamson (1997). This would represent potential further cost savings by reducing the length of the the ensemble and test simulations. We were not dissuaded by the fact that the rapid growth of roundoff order perturbations in CAM5 negatively impacted PerGro's ability to detect changes due to its comparison

with machine epsilon. In fact, we show that examination of ensemble variability after several time steps permits accurate pass and fail determinations and complements CAM-ECT in terms of identifying potential problems. In this paper we present an ensemble-based consistency test that evaluates statistical distinguishability at nine time steps, hereafter designated the CAM Ultra-Fast CAM Ensemble Consistency Test (UF-CAM-ECT).

A notable difference between CAM-ECT and UF-CAM-ECT is the type of data considered. CAM-ECT spatially averages

the yearly mean output to make the ensemble more robust (effectively a double average). Therefore, a limitation of CAM-ECT



is that if a bug only produces a small scale effect, then the overall climate may not be altered in an average sense at 12 months, and the change may go undetected. In this case a longer simulation time may be needed for the bug to impact the average climate. An example of this issue is the modification of the dynamics hyperviscosity parameter (NU) in Baker et al. (2015), which was not detected by CAM-ECT. In contrast, UF-CAM-ECT takes the spatial means of instantaneous values very early

in the model run, which can facilitate the detection of smaller-scale modifications. In terms of simulation length for UF-CAM-ECT, we were aware that we would need to satisfy two constraints in choosing an adequate number of initial time steps: some variables can suffer excessive spread while others remain relatively constant, complicating pass/fail determinations. Balancing the run time and ensemble variability (hence test classification power) also alters the types of statistical differences the test can detect; exploring the complementarity between CAM-ECT and UF-CAM-ECT is a focus of our work.

In particular, we make four contributions: we demonstrate that adequate ensemble variability can be achieved at the ninth CESM time step in spite of the heterogeneous spread among the variables considered; we evaluate UF-CAM-ECT with experiments from Baker et al. (2015), code modifications from Milroy et al. (2016) and several new CAM tests; we propose an effective ensemble size; and we demonstrate that changes to the Community Land Model (CLM) can be detected by both UF-CAM-ECT and CAM-ECT.

In Sect. 2, we provide background information on CESM-ECT, describe our experimental setup and tools, and quantify CESM divergence by time step. In Sect. 3, we detail the UF-CAM-ECT method. We demonstrate the results of our investigation into the appropriate ensemble size in Sect. 4. Experimental results are given in Sect. 5, and we provide guidance for the tools' usage in Sect. 6 and conclude with Sect. 7.

## 2   Background and motivation

### 2.1   CESM and ECT

The CESM is composed of multiple geophysical models (e.g. atmosphere, ocean, land, etc.); the original CESM-ECT work in Baker et al. (2015) focuses on data from the Community Atmosphere Model (CAM). Currently, CESM-ECT consists of tests for CAM (CAM-ECT and UF-CAM-ECT) and POP (POP-ECT).

For CAM-ECT, CAM output data containing annual averages at each grid point for the atmosphere variables are written in
time slices to NetCDF history files. The CAM-ECT ensemble consists of CAM output data from simulations (151 by default or up to 453 simulations in Milroy et al. (2016)) of 1-year in length created on a trusted machine with a trusted version of the CESM software stack. In Baker et al. (2015), the CAM-ECT ensemble is a collection of CESM simulations with identical parameters, differing only in initial atmospheric temperature conditions. In Milroy et al. (2016), the size and composition of the ensemble are further explored, resulting in a refined ensemble comprised of equal numbers of runs from different compilers
(Intel, GNU and PGI), and a recommended size of 300 or 453.

Unique initial temperature perturbations guarantee unique trajectories through the models' phase space, which in turn generate an ensemble of output variables that represents the natural variability in the model. In both Baker et al. (2015) and Milroy et al. (2016), perturbations of the initial atmospheric temperature field are $\mathcal{O}(10^{-14})$, which permits a divergence of the phase





space trajectories, and ensures the statistical properties of the ensemble members are the same at 12 months (Baker et al., 2015).

To compare the statistical characteristics of new runs, CAM-ECT begins by quantifying the variability of the accepted ensemble. Principal Component Analysis (PCA) is used to transform the original space of CAM variables into a subspace of linear combinations of the standardized variables, which are then uncorrelated. The scope of our method is flexible- we can simply add additional output variables (including diagnostic variables) to the test as long as the number of ensemble members is greater than the number of variables, and redo the PCA. To check for consistency with the ensemble, a small set of new runs (three) is fed to the Python CESM Ensemble Consistency Tool (pyCECT), which projects the new runs' outputs into the Principal Component (PC) space of the ensemble. pyCECT issues a pass or fail based on the number of PC scores that fall outside a specified confidence interval (95% by default).

More recently, another module of CESM-ECT was developed to determine statistical consistency for the Parallel Ocean Program (POP) model of CESM and designated POP-ECT (Baker et al., 2016). While similar to CAM-ECT in that an ensemble of trusted simulations is used to evaluate new runs, the statistical consistency test itself is not the same. In POP data, the spatial variation and temporal scales are much different than in CAM data, and there are many fewer variables. Hence the test does not involve PCA or spatial averages, instead making comparisons at each grid location. Note that in this work we demonstrate that CAM-ECT and UF-CAM-ECT testing can be applied directly and successfully to the Community Land Model (CLM) component of CESM, because of the tight coupling between CAM and CLM, indicating that a separate ECT module for CLM is likely unnecessary.

## 2.2 Motivation: CAM divergence in initial time steps

Applying an ensemble consistency test at nine time steps is sensible only if there is an adequate amount of ensemble variability to correctly evaluate new runs as has been shown for the one year runs. This issue is key: with too much spread a bug cannot be detected, and without enough spread the test can be too restrictive in its pass and fail determinations. Many studies consider the effects of initial condition perturbations to ensemble members on the predictability of an ESM, and the references in Kay et al. (2015) contain several examples. In particular, Deser et al. (2012) studies uncertainty arising from climate model internal variability using an ensemble method, and an earlier work (Branstator and Teng, 2010) considers the predictability and forecast range of a climate model by examining the separate effects and time scales of initial conditions and forcings. Branstator and Teng (2010) also study ensemble global means and spread, as well as undertaking an entropy analysis of leading Empirical Orthogonal Functions (comparable to PCA). These studies are primarily concerned with model variability and predictability at the timescale of a several years or more. However, we note that concurrent to our work, a new method that considers one second time steps has been developed in Wan et al. (2017). Their focus is on comparing the numerical error in time integration between a new run and control runs.

As mentioned previously, we were curious about the behavior of CESM in its initial time steps in terms of whether we would be able to determine statistical distinguishability. Fig. 1 represents our initial inquiry into this behavior. To generate the data, we ran two simulations of 11 time steps each: one with no initial condition perturbation and one with a perturbation of $\mathcal{O}(10^{-14})$



to the initial atmospheric temperature. The vertical axis labels designate CAM variables, while the horizontal axis specifies the CESM time step. The color of each step represents the number of significant figures in common between the perturbed and unperturbed simulations' area weighted global means: a small number of figures in common (darker red) indicates a large difference. Black tiles specify time steps where the variable's value is not computed due to model sub-cycling (Hurrell et al.,

2013). White tiles indicate between 10 and 17 significant figures in common (i.e., a small magnitude of difference). Most CAM variables exhibit a difference from the unperturbed simulation at the initial time step (0), and nearly all have diverged to only a few figures in common by step 10. This observation suggests that we can create an ensemble possessing sufficient variability to determine statistical distinguishability using a small number of time steps. We further examine the ninth time step as it is the last step on the plot where sub-cycled variables are calculated. Of the total 134 CAM variables output by default in our

version of CESM (see Sect. 3), 117 are utilized by CAM-ECT, as 17 are either redundant or have zero variance. In all following analyses, the first nine sub-cycled variables distinguished by red labels (AODDUST1, AODDUST3, AODVIS, BURDENBC, BURDENDUST, BURDENPOM, BURDENSEASALT, BURDENSO4, and BURDENSOA) are also discarded, as they take constant values through time step 45. Thus we use 108 variables from Fig. 1 in the UF-CAM-ECT ensemble.

Next we examine the time series of each CAM variable by looking at the first 45 CESM time steps ($t_0$ through $t_{45}$) for 30

simulations to select a time step when all variables experience sufficient divergence from the values of the reference unperturbed simulation. To illustrate ensemble variability at initial time steps, Fig. 2 depicts the time evolution of three representative CAM variables from $t_0$ to $t_{45}$. Most CAM variables' behavior is analogous to one of the rows in this figure. The dataset was generated by running 31 simulations: one with no initial atmospheric temperature perturbation, and 30 with different $\mathcal{O}(10^{-14})$K perturbations. The vertical axis labels the difference between the unperturbed simulation and the perturbed simulations' area

weighted global means, divided by the unperturbed simulation's area weighted global mean value for the indicated variable at each time step. The right column visualizes the distributions of the data in the left column. Each box plot represents the values in the left column at nine time step intervals from 9-45 (inclusive). In most cases, the variables attain measurable but well-contained spread in approximately the first nine time steps. From the standpoint of CAM-ECT, this means that an ensemble created at this step will likely contain sufficient variability to categorize experimental sets correctly. While nine time steps

may not be strictly optimal, we have no reason to believe that more time steps results in a more accurate ensemble consistency determination. We further discuss this topic in Sect. 4 with an investigation of the properties of ensembles created from the ninth time step and compare their pass/fail determinations of experimental simulations with that of CAM-ECT.

## 3   UF-CAM-ECT approach

UF-CAM-ECT employs the same essential test method as CAM-ECT described in Baker et al. (2015), but with a CESM

simulation length of nine time steps (which is approximately 5 simulation hours) using the default CAM time step of 1800 seconds (30 minutes). By considering a specific time step, we are using instantaneous values in contrast to CAM-ECT, which uses yearly average values. UF-CAM-ECT inputs are spatially averaged, so averaged once, whereas CAM-ECT inputs are averaged across the 12 simulation months and spatially averaged, so averaged twice. As a consequence of using instantaneous





values, UF-CAM-ECT is more sensitive to localized phenomena (see Sect. 5.2). By virtue of the small number of modifications required to transform CAM-ECT into UF-CAM-ECT, we consider the ECT framework to have surprisingly broad applicability. Substituting instantaneous values for yearly averages permits the discernment of different features and modifications- see Sects. 5 and 5.2 for evidence of this assertion.

As in Baker et al. (2015) and Milroy et al. (2016), we generate the CESM results on a 1° global grid using the CAM5 model version described in Kay et al. (2015), and despite the rapid growth in perturbations in CAM5 with the default time step of 1800 seconds, we can still characterize its variability. We run simulations with 900 MPI tasks and two OpenMP threads per task (unless otherwise noted) on the Yellowstone machine at NCAR. The iDataPlex cluster is composed of 4,536 compute nodes, featuring two Xeon Sandy Bridge CPUs with 32 GB memory and one FDR InfiniBand interconnect per node. The

default compiler on Yellowstone for our CESM version is Intel 13.1.2 with -O2 optimization; GNU 4.8.0 and PGI 13.0 are also CESM-supported compilers that are used in this study. With 900 MPI processes and two OpenMP threads per process, a simulation of nine time steps on Yellowstone is a factor of approximately 70 cheaper in terms of CPU time than a one year CESM simulation.

Either single or double precision output is suitable for UF-CAM-ECT. While CAM can be instructed to write its history
files in single or double precision floating point form, its default is single precision which was used for CAM-ECT in Baker et al. (2015) and Milroy et al. (2016). Similarly, UF-CAM-ECT takes single precision output by default. However, we chose to generate double precision output to facilitate the study represented by Fig. 1; it would have been impossible to perform a significance test of up to 17 digits otherwise. In the case of new runs written in double precision, both CAM-ECT and UF-CAM-ECT compare ensemble values promoted to double precision with the unmodified new outputs. We determined that the
effects of using double or single precision outputs for ensemble generation and the evaluation of new runs did not impact statistical distinguishability.

## 4    UF-CAM-ECT ensemble size

In this section we consider the properties of the UF-CAM-ECT ensemble, particularly focusing on ensemble size. Given the use of instantaneous values at nine time steps in UF-CAM-ECT, our expectation was that the size of the ensemble would
differ from that of CAM-ECT. We considered it plausible that a larger number would be required to make proper pass and fail determinations. The ensemble should contain enough variability that UF-CAM-ECT classifies experiments expected to be statistically indistinguishable as consistent with the ensemble. Furthermore, for experiments that significantly alter the climate, UF-CAM-ECT should classify them as statistically distinct from the ensemble. Accordingly, the ensemble itself is key, and examining its size allows us to quantify the variability it contains as the number of ensemble members increases.

Our sets of experimental simulations (new runs) typically consist of 30 members, but by default pyCECT was written to do a single test on three runs. Performing the full set of possible CAM-ECT tests from a given set of experimental simulations allows us to make robust failure determinations as opposed to a single binary Pass/Fail test. In this work we utilize the Ensemble Exhaustive Test (EET) tool described in Milroy et al. (2016) to calculate an overall failure rate for sets of new runs larger than





the pyCECT default. A failure percent provides more detail on the statistical difference between the ensemble and experimental set. To calculate the failure rate, EET efficiently performs all possible tests which are equal in number to the ways $N_{test}$ simulations can be chosen from all $N_{tot}$ simulations (i.e., the binomial coefficient: $\binom{N_{tot}}{N_{test}}$). For this work most experiments consist of 30 simulations, so $N_{tot} = 30$ and $N_{test} = 3$ yields 4,060 possible combinations. With this tool we can make a

comparison between the exhaustive test failure rate and the single-test CAM-ECT false positive rate calibrated to be 0.5%.

To determine a desirable UF-CAM-ECT ensemble size, we gauge whether ensembles of varying sizes contain sufficient variability by holding out sets of ensemble simulations and performing exhaustive testing against ensembles formed from the remaining elements. Since the test sets and ensemble members are generated by the same type of initial condition perturbation, the tests should pass. We begin with a set of 801 CESM simulations of nine time steps, differing by unique perturbations

to the initial atmospheric temperature field in $\{\left[-9.99 \times 10^{-14}, 0\right), \left(0, 9.99 \times 10^{-14}\right]\}$ K. The motivation for generating a large number of outputs was our expectation that ensembles created from instantaneous values would contain less variability. Moreover, since the runs are comparatively cheap, it was easy to run many simulations for testing purposes. We first randomly select a subset from the 801 simulations and compute the PC loadings. From the remaining simulations, we choose 30 at random and run EET against this experimental set. For each ensemble size, we make 100 random draws to form an ensemble,

and for each ensemble we make 100 random draws of experimental sets. This results in 10,000 EET results per ensemble size. For example, to test the variability of the size 350 ensemble, we choose 350 simulations at random from our set of 801 to form an ensemble. From the remaining 451 simulations, we randomly choose 30 and exhaustively test them against the generated ensemble with EET (4,060 individual tests). This is repeated 99 times for the ensemble. Then 99 more ensembles are created in the same way, yielding 10,000 tests for size 350. As such, we tested sizes from 100 through 750, and include a plot of the

results in Fig. 3. Since all 801 simulations are created by the same type of perturbation, we expect EET to issue a pass for each experimental set against each ensemble. This study is essentially a bootstrap method used jointly with cross validation to ascertain the minimum ensemble size for stable PC calculations and pass/fail determinations. With greater ensemble size the distribution of EET failure rates should narrow, reflecting the increased stability of calculated PC loadings that accompanies larger sample sizes. The EET failure rates will never be uniformly zero due to the statistical nature of the test. The chosen false

positive rate of 0.5% is reflected by the red horizontal line in Fig. 3. We define an adequate ensemble size as one whose median is less than 0.5% and whose interquartile range (IQR) is narrow. The IQR is defined as the difference between the upper and lower quartiles of a distribution. For the remainder of this work we use the size 350 ensemble shown in Fig. 3, as it is the smallest ensemble that meets our criteria of median below 0.5% and narrow IQR. The larger ensembles represent diminishing returns at greater computational expense.

**5   Results**

The UF-CAM-ECT must have properties comparable or complementary to CAM-ECT including high classification accuracy. In particular, its response to modifications known to produce statistically distinguishable output should be a fail, and to changes not expected to result in statistically distinguishable output, a pass. We verify its effectiveness by performing the same tests as





before with CAM-ECT: CAM namelist alterations and compiler changes from Baker et al. (2015) as well as code modifications from Milroy et al. (2016). We further explore UF-CAM-ECT properties with experiments from CLM and several new CAM experiments. In the following sections, UF-CAM-ECT experiments consist of 30 runs due to their low cost, allowing us to do exhaustive testing. For CAM-ECT, we only run EET (which is far more expensive due to the need for more than three

12-month runs) in Sect. 5.3, where the expected experiment outcomes are less certain. The UF-CAM-ECT ensemble selected for testing is of size 350 (see Fig. 3), and the CAM-ECT ensemble is size 300, comprised of 100 simulations built by Intel, GNU, and PGI compilers (the smallest size recommended in Milroy et al. (2016)).

## 5.1 Matching expectation and result: where UF and CAM-ECT agree

UF-CAM-ECT should return a pass when run against experiments expected to be statistically indistinguishable from the ensem-
ble. Examples of this category include building CESM with a different compiler or a different floating point results preserving optimization order (e.g., with no optimization: O0), running CESM without OpenMP threading or on other CESM-supported platforms, and minor code modifications. We present results of the following tests in this category that should pass from Baker et al. (2015): *NO-OPT*, *INTEL-15*, *NO-THRD*, *PGI*, *GNU*, and *EDISON* (see Appendix A1 for descriptions). From Milroy et al. (2016), we present results of tests with minimal code modifications that were developed to test the variability and classification
power of CESM-ECT. These code modifications should also pass UF-CAM-ECT: *Combine* (C), *Expand* (E), *Division-to-Multiplication* (DM), *Unpack-Order* (UO), and *Precision* (P) (see Appendix A2 for full descriptions). A comparison between the EET failures of UF-CAM-ECT and the single test CAM-ECT for these experiments that should all pass is presented in the upper section of Table 1. Note that all EET failure rates for these experiments are $< 1\%$ for UF-CAM-ECT, indicating full agreement between CAM-ECT and UF-CAM-ECT.

20       To further exercise UF-CAM-ECT, we perform the CAM namelist experiments expected to fail from Baker et al. (2015): *DUST*, *FACTB*, *FACTIC*, *RH-MIN-LOW*, *RH-MIN-HIGH*, *CLDFRC-DP*, *UW-SH*, *CONV-LND*, *CONV-OCN*, and *NU-P* (see A1 for descriptions). These "expected to fail" results are presented in the lower section of Table 1. For UF-CAM-ECT each result in the table is identically a 100% EET failure: a clear indication of statistical distinctness from the size 350 UF-CAM-ECT ensemble. For all of the examples presented in Table 1 the CAM-ECT and UF-CAM-ECT tests are in agreement, which
is a testament to the utility of UF-CAM-ECT.

## 5.2 CLM Modifications

The Community Land Model (CLM), the land model component of CESM, was initially developed to study land surface processes and land-atmosphere interactions, and was a product of a merging of a community land model with the NCAR Land Surface Model (Oleson et al., 2010). More recent versions benefit from the integration of far more sophisticated phys-
ical processes than in the original code. Specifically, CLM 4.0 integrates models of vegetation phenology, surface albedos, radiative fluxes, soil and snow temperatures, hydrology, photosynthesis, river transport, urban areas, carbon-nitrogen cycles, and dynamic global vegetation, among many others (Oleson et al., 2010). Moreover, the CLM receives state variables from CAM and updates hydrology calculations, outputting the fields back to CAM (Oleson et al., 2010). It is sensible to assume that



since information propagates between the land and atmosphere models, in particular between CLM and CAM, CAM-ECT and UF-CAM-ECT would be capable of detecting changes to CLM.

For our tests we use CLM version 4.0, which is the default for our CESM version (see Sect. 3) and the same version used in all experiments in this work. Our CLM experiments are described as follows:

– **CLM_INIT** changes from using the default land initial condition file to using a cold restart.

– **CO2_PPMV_280** reduces the $CO_2$ type and concentration from CLM_CO2_TYPE = 'diagnostic' to CLM_CO2_TYPE = 'constant' and CCSM_CO2_PPMV = 280.0.

– **CLM_VEG** activates CN mode (carbon-nitrogen cycle coupling).

– **CLM_URBAN** disables urban air conditioning/heating and the wasteheat associated with these processes so that the
internal building temperature floats freely.

See Table 2 for the test results of the experiments. The pass and fail results in this table reflect our high confidence in the expected outcome: all test determinations are in agreement, the UF-CAM-ECT passes represent EET failure rates $< 1\%$, and failing UF-CAM-ECT tests are all 100% EET failures. We expected failures for CLM_INIT because the CLM and CAM coupling period is 30 simulation minutes, and such a substantial change to the initial conditions should be detected immediately
and persist through 12 months. CLM_CO2_PPMV_280 is also a tremendous change as it effectively resets the atmospheric $CO_2$ concentration to a preindustrial value, and changes which $CO_2$ value the model uses. In particular, for CLM_CO2_TYPE = 'diagnostic' CLM uses the value from the atmosphere (367.0), while CLM_CO2_TYPE = 'constant' instructs CLM to use the value specified by CCSM_CO2_PPMV. Therefore both tests detect the large reduction in $CO_2$ concentration, generating failures at the ninth time step and in the 12 month average. CLM_VEG was also expected to fail immediately, given how
quickly the CN coupling is expressed. Finally, the passing results of both CAM-ECT and UF-CAM-ECT for CLM_URBAN is unsurprising as the urban fraction is less than 1% of the land surface, and heating and air conditioning only occur over a fraction of this 1% as well.

Our experiments thus far indicate that both CAM-ECT and UF-CAM-ECT will detect errors in CLM, and that a separate CESM-ECT module for CLM (required for POP) is most likely not needed. While this finding may be unsurprising given
how tightly CAM and CLM are coupled, it represents a significant broadening of the tools' applicability and utility. Note that while we have not generated CAM-ECT or UF-CAM-ECT ensembles with CN mode activated in CLM (which is a common configuration for land modeling), we have no reason to believe that statistical consistency testing of CN-related CLM code changes would not be equally successful. Consistency testing of active CN mode code changes bears further investigation and will be a subject of future work.

## 5.3   UF-CAM-ECT and CAM-ECT disagreement

In this section we test experiments that result in contradictory determinations by UF-CAM-ECT and CAM-ECT. Due to the disagreement, all tests' EET failure percentages are reported for 30 run experimental sets for both UF-CAM-ECT and CAM-



ECT. We present the results in Table 3. The modifications are described in the following list (note that *NU* and *RAND-MT* can also be found in Baker et al. (2015) and Milroy (2015), respectively):

- *RAND-MT* substitutes the Mersenne Twister Pseudo Random Number Generator (PRNG) for the default PRNG in radiation modules.

- *TSTEP_TYPE* changes the time stepping method for the spectral element dynamical core from 4 (Kinnmark & Gray Runga-Kutta 4 stage) to 5 (Kinnmark & Gray Runga-Kutta 5 stage).

- *QSPLIT* alters how often tracer advection is done in terms of dynamics time steps, the default is 1, and we increase it to 9.

- *CPL_BUG* sets albedos to zero above 57 degrees North latitude in the coupler.

- *CLM_HYDRO_BASEFLOW* increases the soil hydrology baseflow rate coefficient in CLM from $5.5 \times 10^{-3}$ to 55.

- *NU* changes the dynamics hyperviscosity (horizontal diffusion) from $1 \times 10^{15}$ to $9 \times 10^{14}$.

- *CLM_ALBICE_00* changes the albedo of bare ice on glaciers (visible and near-infrared albedos for glacier ice) from 0.80,0.55 to 0.00,0.00.

### 5.3.1 Minor setting changes: RAND-MT, TSTEP_TYPE, and QSPLIT

RAND-MT is a test of the response to substituting the CAM default Pseudo Random Number Generator (PRNG) in the radiation module for a different CESM-supported PRNG (Milroy, 2015). Since the PRNG affects radiation modules which compute cloud properties, it is reasonable to conclude that the change alters the distributions of cloud-related CAM variables (such as cloud covers). Both CAM and its PRNG are deterministic; the variability at nine time steps exhibits different characteristics depending on the PRNG. However, we would not expect (nor would we want) a change to the PRNG to induce statistically distinguishable results over a longer period such as a simulation year, and this expectation is confirmed by CAM-ECT.

TSTEP_TYPE and QSPLIT are changes to attributes of the model dynamics: TSTEP_TYPE alters the time stepping method in the dynamical core, and QSPLIT modifies the frequency of tracer advection computation relative to the dynamics time step. It is well known that CAM is generally much more sensitive to the physics time step than to the dynamics time step. Time-stepping errors in CAM dynamics do not affect large-scale well-resolved waves in the atmosphere but they do affect small-scale fast waves. While short-term weather should be affected, the model climate is not expected to be affected by time-stepping method or dynamics time-step. However, like the RAND-MT example, UF-CAM-ECT registers the less "smoothed" instantaneous global means as failures for both tests, while CAM-ECT finds the yearly averaged global means to be statistically indistinguishable. Small grid-scale waves are affected by choice of TSTEP_TYPE in short runs, however, the long-term climate is not affected by time-stepping method. The results of these experiments are shown in the top section of Table 3, and for experiments of this type, CAM-ECT yields anticipated results. This group of experiments exemplifies the categories of experiments to which UF-CAM-ECT may be sensitive: small-scale or minor changes to initial conditions or settings which are





irrelevant in the long term. Therefore, while the UF-CAM-ECT results can be misleading in particular in these cases, they may indicate a larger problem as will be seen in the examples in Sect. 5.3.3.

### 5.3.2 Contrived experiments: CPL_BUG and CLM_HYDRO_BASEFLOW

Motivated by experiments which bifurcate the tests' findings, we seek the reverse of the previous three experiments in Sect. 5.3.1: examples of a parameter change or code modification that are distinguishable in the yearly global means, but are undetectable in the first time steps. We consulted with climate scientists and CESM software engineers, testing a large number of possible modifications to find some that would pass UF-CAM-ECT and fail CAM-ECT. The results in the center section of Table 3 represent a small fraction of the tests performed, as examples that met the condition of UF-CAM-ECT pass and CAM-ECT fail were exceedingly difficult to find. In fact, CPL_BUG and CLM_HYDRO_BASEFLOW were devised specifically for that purpose. That their failure rates are far from 100% is an indication of the challenge of finding an error that is not present at nine time steps, but manifests clearly in the annual average.

CPL_BUG is devised to demonstrate that it is possible to construct an example that does not yield substantial differences in output at nine time steps, but does impact the yearly average. Selectively setting the albedos to zero above 57 degrees latitude has little effect at nine time steps since this region experiences almost zero solar radiation during the first five hours of January 1. The nonzero CAM-ECT result is a consequence of using annual averages since for the Northern hemisphere summer months this region is exposed to nearly constant solar irradiance.

CLM_HYDRO_BASEFLOW is another manufactured example of a change designed to be undetectable at the ninth time step. It is an increase in the exponent of the soil hydrology baseflow rate coefficient, which controls the amount of water drained from the soil. This substantial change (four orders of magnitude) cannot be detected by UF-CAM-ECT since the differences at nine time steps are confined to deep layers of the soil. However, through the year they propagate to and eventually influence the atmosphere through changes in surface fluxes, which is corroborated by the much higher CAM-ECT failure rate.

### 5.3.3 Small but consequential changes: NU and CLM_ALBICE_00

CAM-ECT results in Baker et al. (2015) for the NU experiment are of particular interest as climate scientists expected this experiment to fail. NU is an extraordinary case, as Baker et al. (2015) acknowledge: "Because CESM-ECT [currently CAM-ECT] looks at variable annual global means, the 'pass' result [for NU] is not entirely surprising as errors in small-scale behavior are unlikely to be detected in a yearly global mean." The change to NU should be evident only where there are strong field gradients and small scale precipitation. We applied EET for CAM-ECT with 30 runs and determined the failure rate to be 33.0% against the reference ensemble. In terms of CAM-ECT this experiment was borderline, as the probability that CAM-ECT will classify three NU runs a "pass" is not much greater than a "fail" outcome. In contrast UF-CAM-ECT is able to detect this difference much more definitively in the instantaneous data at the ninth time step. We would also expect this experiment to fail more definitively for simulations longer than 12 months, once the small but nevertheless consequential change in NU had time to manifest.





CLM_ALBICE_00 affects a very small percent of the land area. Furthermore, of that land area, only regions where the fractional snow cover is $< 1$ and incoming solar radiation is present will be affected by the modification to the bare ice albedo. Therefore, it was expected to pass both CAM-ECT and UF-CAM-ECT, yet the EET failure rate for UF-CAM-ECT was 96.3%. We consider the CLM_ALBICE_00 experiment in greater detail to better understand the differences between

UF-CAM-ECT and CAM-ECT. Since the change is small and localized, we need to discover the reason why UF-CAM-ECT detects a statistical difference, particularly given that many northern regions with glaciation receive little or no solar radiation at time step 9 (January 1). To explain this unanticipated result, we created box plots of all 108 CAM variables tested by UF-CAM-ECT to compare the distributions (at nine time steps and at one year) of the ensemble versus the 30 CLM_ALBICE_00 simulations. Each plot was generated by subtracting the unperturbed ensemble value from each value, then rescaling by the

unperturbed ensemble value. After analyzing the plots, we isolated four variables (FSDSC: clearsky downwelling solar flux at surface, FSNSC: clearsky net solar flux at surface, FSNTC: clearsky net solar flux at top of model, and FSNTOAC: clearsky net solar flux at top of atmosphere) that exhibited markedly different behaviors between the ensemble and experimental outputs. Fig. 4 displays the results. The left column represents distributions from the ninth time step which demonstrate the distinction between the ensemble and experiment: the top three variables' distributions have no overlap. For the 12 month runs, the

ensemble and experiments are much less distinct. It is sensible that the global mean net fluxes are increased by the albedo change, as the incident solar radiation should be a constant, while the zero albedo forces all radiation impinging on exposed ice to be absorbed. The absorption reduces the negative radiation flux, making the net flux more positive. The yearly mean distributions are not altered enough for CAM-ECT to robustly detect a fail (12.8% EET failure rate), which is due to feedback mechanisms having taken hold and leading to spatially heterogeneous effects, which are seen as such in the spatial and temporal

12 month average.

The percentage of grid cells affected by the CLM_ALBICE_00 modification is 0.36% (calculated by counting the number of cells with nonzero FSNS, fractional snow cover (FSNO in CLM) $\in (1, 0)$, and PCT_GLACIER greater than zero in the surface dataset). Remarkably, despite such a small area being affected by the albice change, UF-CAM-ECT flags these simulations as statistically distinguishable. The results of CLM_ALBICE_00 taken together with NU indicate that UF-CAM-ECT demon-

strates the ability to detect small-scale events, in fulfillment of the desired capability for CAM-ECT mentioned as future work in Baker et al. (2015).

## 6  Implications and ECT Guidelines

In this section we summarize the lessons learned in Sect. 5 to provide both clarification and guidance on the use of the complementary tools UF-CAM-ECT and CAM-ECT in practice. Our extensive experiments, a representative subset of which are

presented in Sect. 5, indicate that UF-CAM-ECT and CAM-ECT typically return the same determination. Indeed, finding counterexamples was non-trivial. Certainly the types of modifications that occur frequently in the CESM development cycle (e.g., compiler upgrades, new CESM-supported platforms, minor code rearrangements for optimization, and initial state changes) are all equally well classified by both UF-CAM-ECT and CAM-ECT. Therefore, in practice we recommend the use




of the cheaper UF-CAM-ECT as a first step for port verification, code optimization and compiler flag changes, as well as other frequent CESM quality assurance procedures.

CAM-ECT is used as a second step only when needed for complementary information as follows. First, our experimentation indicates that if UF-CAM-ECT issues a pass, it is very likely that CAM-ECT will also issue a pass. While discovering examples

where UF-CAM-ECT issues a pass and CAM-ECT issues a fail is theoretically possible (e.g. a seasonal or slow-propagating effect), in practice the two examples we gave in Sect. 5.3.2 were quite contrived as we could not identify more realistic ones. It appears that if a change propagates so slowly as not to be detected at the ninth time step, its later effects can be smoothed by the annual averaging which includes the initial behavior. Accordingly, the change may go undetected by CAM-ECT when used without EET (e.g., failure rates for CAM-ECT in the lower third of Table 3 are well below 100%). Therefore in practice,

applying CAM-ECT as a second step should only be considered when UF-CAM-ECT issues a fail. In particular, because we have shown that UF-CAM-ECT is quite sensitive to small-scale errors or alterations (see CLM_ALBICE_00 in Sect. 5.3.3 which impacted less than 1% of land area), by running CAM-ECT when UF-CAM-ECT fails, we can further determine whether the change also impacted statistical consistency during the first year. If CAM-ECT also fails then the UF-CAM-ECT result is confirmed. On the other hand, if CAM-ECT passes, the situation is more nuanced. Either a small-scale change has occurred

that is unimportant long-term for the mean climate (e.g., RAND-MT), or a small-scale change has occurred that will require a longer time scale than 12 months to manifest decisively (e.g., NU). In either case, the user must have an understanding of the characteristics of the modification being tested to reconcile the results at this point. Future work will include investigation of ensembles at longer time scales, which will aid in the overall determination of the relevance of the error.

## 7   Conclusions

We developed a new Ultra-Fast CAM Ensemble Consistency test from output at the ninth time step of the CESM. Conceived largely out of curiosity, it proved to possess surprisingly wide applicability in part due to its use of instantaneous values rather than annual means. The short simulation time translated to a cost savings of a factor of approximately 70 over a simulation of 12 months, considerably reducing the expense of ensemble and test run creation. Through methodical testing, we selected a UF-CAM-ECT ensemble size (350) that balances the variability contained in the ensemble (hence its ability to classify new

runs) with the cost of generation. We performed extensive experimentation to test which modifications known to produce statistically distinguishable and indistinguishable results would be classified as such by UF-CAM-ECT. These experiments yielded clear pass/fail results that were in agreement between the two tests, allowing us to more confidently prescribe use cases for UF-CAM-ECT. Due to the established feedback mechanisms between the Community Land Model (CLM) component of CESM and CAM, we extended CESM-ECT testing to CLM. Our determination that both CAM-ECT and UF-CAM-ECT are

capable of identifying statistical distinguishability resulting from alterations to CLM indicates that a separate ECT module for CLM is likely unnecessary. By studying experiments where CAM-ECT and UF-CAM-ECT arrived at different findings we concluded that UF-CAM-ECT is capable of detecting small-scale changes, a feature that facilitates root cause analysis for test failures in conjunction with CAM-ECT.



UF-CAM-ECT will be an asset to CESM model developers, software engineers, and climate scientists. The ultra-fast test is cheap and quick, and further testing is not required when a passing result indicating statistical consistency is issued. Ultimately the two tests can be used in concert to provide richer feedback to software engineers, hardware experts, and climate scientists: combining the results from the ninth time step and 12 months enhances understanding of the time scales on which changes be-

come operative and influential. We intend to refine our understanding of both UF-CAM-ECT and CAM-ECT via an upcoming study on decadal simulations. We hope to determine whether the tests are capable of identifying statistical consistency (or lack thereof) of modifications that may take many years to manifest fully. Another potential application of the tests is the detection of hardware or software issues during the initial evaluation and routine operation of a supercomputer.

## 8   Code and Data Availability

The current version (v3.0.0) of Python tools can be obtained directly from https://github.com/NCAR/PyCECT. In addition, CESM-ECT is available as part of the Common Infrastructure for Modeling Earth (CIME) at https://github.com/ESMCI/cime. CESM-ECT will be included in the CESM public releases beginning with the version 2.0 series (expected to be available in Spring 2017).

### Appendix A: Referenced experiments

**A1   Experiments from Baker et al. (2015)**

- NO-OPT: changing the Intel compiler flag to remove optimization

- INTEL-15: changing the Intel compiler version to 15.0.0

- NO-THRD: compiling CAM without threading (MPI-only)

- PGI: using the CESM-supported PGI compiler (13.0)

- GNU: using the CESM-supported GNU compiler (4.8.0)

- EDISON: National Energy Research Scientific Computing Center (Cray XC30, Intel)

- DUST: dust emissions; dust_emis_fact = 0.45 (original default 0.55)

- FACTB: wet deposition of aerosols convection factor; sol_factb_interstitial = 1.0 (original default 0.1)

- FACTIC: wet deposition of aerosols convection factor; sol_factic_interstitial = 1.0 (original default 0.4)

- RH-MIN-LOW: min. relative humidity for low clouds; cldfrc_rhminl = 0.85 (original default 0.8975)

- RH-MIN-HIGH: min. relative humidity for high clouds; cldfrc_rhminh = 0.9 (original default 0.8)





- CLDFRC-DP: deep convection cloud fraction; cld_frc_dp1 = 0.14 (original default 0.10)

- UW-SH: penetrative entrainment efficiency - shallow; uwschu_rpen = 10.0 (original default 5.0)

- CONV-LND: autoconversion over land in deep convection; zmconv_c0_lnd = 0.0035 (original default 0.0059)

- CONV-OCN: autoconversion over ocean in deep convection; zmconv_c0_ocn = 0.0035 (original default 0.045)

- NU-P: hyperviscosity for layer thickness (vertical lagrangian dynamics); nu_p = $1 \times 10^{14}$ (original default $1 \times 10^{15}$)

- NU: dynamics hyperviscosity (horizontal diffusion); nu = $9 \times 10^{14}$ (original default $1 \times 10^{15}$)

## A2    Experiments from Milroy et al. (2016)

- ***Combine*** (**C**) is a single line code change to the *preq_omega_ps* subroutine.

- ***Expand*** (**E**) is a modification to the *preq_hydrostatic* subroutine. We expand the calculation of the variable `phi`.

- ***Division-to-multiplication*** (**DM**): The original version of the *euler_step* subroutine of the primitive trace advection module (*prim_advection_mod.F90*) includes an operation that divides by a spherical mass matrix `spheremp`. The modification to this kernel consists of declaring a temporary variable (`tmpsphere`) defined as the inverse of `spheremp`, and substituting a multiplication for the more expensive division operation.

- ***Unpack-order*** (**UO**) changes the order that an MPI receive buffer is unpacked in the *edgeVunpack* subroutine of *edge_mod.F90*.
Changing the order of buffer unpacking has implications for performance, as traversing the buffer sub-optimally can prevent cache prefetching.

- ***Precision*** (**P**) is a performance-oriented modification to the water vapor saturation module (*wv_sat_methods.F90*). From a performance perspective this could be extremely advantageous and could present an opportunity for coprocessor acceleration due to superior single-precision computation speed. We modify the elemental function that computes saturation
vapor pressure by substituting `r4` for `r8` and casting to single-precision in the original.

*Acknowledgements.* Many thanks are due Haiying Xu for her help modifying pyCECT for comparison of CESM-ECT single and double precision floating point statistical consistency results. We wish to thank Michael Levy for his suggestion of altering albedos above a certain northern latitude (CPL_BUG), Peter Lauritzen for his fruitful discussion of CAM physics and dynamics, and Keith Oleson for his considerable assistance interpreting and visualizing CLM experiments and output.

We would like to acknowledge high-performance computing support from Yellowstone (ark:/85065/d7wd3xhc) provided by NCAR's Computational and Information Systems Laboratory, sponsored by the National Science Foundation. This research also used computing resources provided by the National Energy Research Scientific Computing Center, a DOE Office of Science User Facility supported by the Office of Science of the US Department of Energy under contract no. DEAC02-05CH11231. This work was funded in part by the Intel Parallel Computing Center for Weather and Climate Simulation (https://software.intel.com/en-us/articles/intel-parallel-computing-center-at-
the-university-of-colorado-boulder-and-the-national).



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





**Table 1.** Experiments where CAM-ECT and UF-CAM-ECT are in agreement from Baker et al. (2015) and Milroy et al. (2016). The CAM-ECT column is the result of a single test on three runs; The UF-CAM-ECT column represents EET results: all Passes are failure rates $< 1\%$, and Fails are EET failure rates equal to $100\%$. See Appendix A1 and Appendix A2 for full descriptions of the experiments.

| Experiment | CAM-ECT results | UF-CAM-ECT results |
| --- | --- | --- |
| NO-OPT | Pass | Pass |
| INTEL-15 | Pass | Pass |
| NO-THRD | Pass | Pass |
| PGI | Pass | Pass |
| GNU | Pass | Pass |
| EDISON | Pass | Pass |
| C | Pass | Pass |
| E | Pass | Pass |
| DM | Pass | Pass |
| UO | Pass | Pass |
| P | Pass | Pass |
| DUST | Fail | Fail |
| FACTB | Fail | Fail |
| FACTIC | Fail | Fail |
| RH-MIN-LOW | Fail | Fail |
| RH-MIN-HIGH | Fail | Fail |
| CLDFRC-DP | Fail | Fail |
| UW-SH | Fail | Fail |
| CONV-LND | Fail | Fail |
| CONV-OCN | Fail | Fail |
| NU-P | Fail | Fail |



**Table 2.** Modifications to code sections of CLM. The CAM-ECT column is the result of a single ECT test on three runs. The UF-CAM-ECT column represents EET failure rates: a pass is $< 1\%$ and a fail is 100%.

| Experiment | CAM-ECT results | UF-CAM-ECT results |
|---|---|---|
| CLM_INIT | Fail | Fail |
| CLM_CO2_PPMV_280 | Fail | Fail |
| CLM_VEG | Fail | Fail |
| CLM_URBAN | Pass | Pass |

**Table 3.** These experiments represent disagreement between UF-CAM-ECT and fail CAM-ECT. Shown are the EET failure rates of 30 run sets.

| Experiment | CAM-ECT failure rates | UF-CAM-ECT failure rates |
|---|---|---|
| RAND-MT | 4.7% | 99.4% |
| TSTEP_TYPE | 2.5% | 100% |
| QSPLIT | 1.8% | 100% |
| CPL_BUG | 41.6% | 0.1% |
| CLM_HYDRO_BASEFLOW | 30.7% | 0.1% |
| NU | 33.0% | 100% |
| CLM_ALBICE_00 | 12.8% | 96.3% |





**Figure 1.** Representation of effects of initial CAM temperature perturbation over 11 time steps (including $t = 0$). CAM variables are listed on the vertical axis, and the horizontal axis records the simulation time step. The color bar designates equality of the corresponding variables between the unperturbed and perturbed simulations' area weighted global means after being rounded to $n$ significant digits ($n$ is the color) at each time step. Time steps where the corresponding variable was not computed (subcycled variables) are colored black. White indicates equality of greater than 9 significant digits (i.e. 10-17). Red variable names are not used by UF-CAM-ECT.





**Figure 2.** The vertical axis labels the difference between the unperturbed simulation and the perturbed simulations' area weighted global means, divided by the unperturbed simulation's area weighted global mean for the indicated variable at each time step. The horizontal axis is the CESM time step with intervals chosen as multiples of nine. The left column plots are time series representations of the values three CAM variables chosen as representatives of the entire set. (Variables behave similarly to one of these three.) The right column plots are statistical representations of the 30 values plotted at each vertical grid line of the corresponding left column. More directly, each box plot depicts the distribution of values of each variable for each time step from 9-45 in multiples of 9.




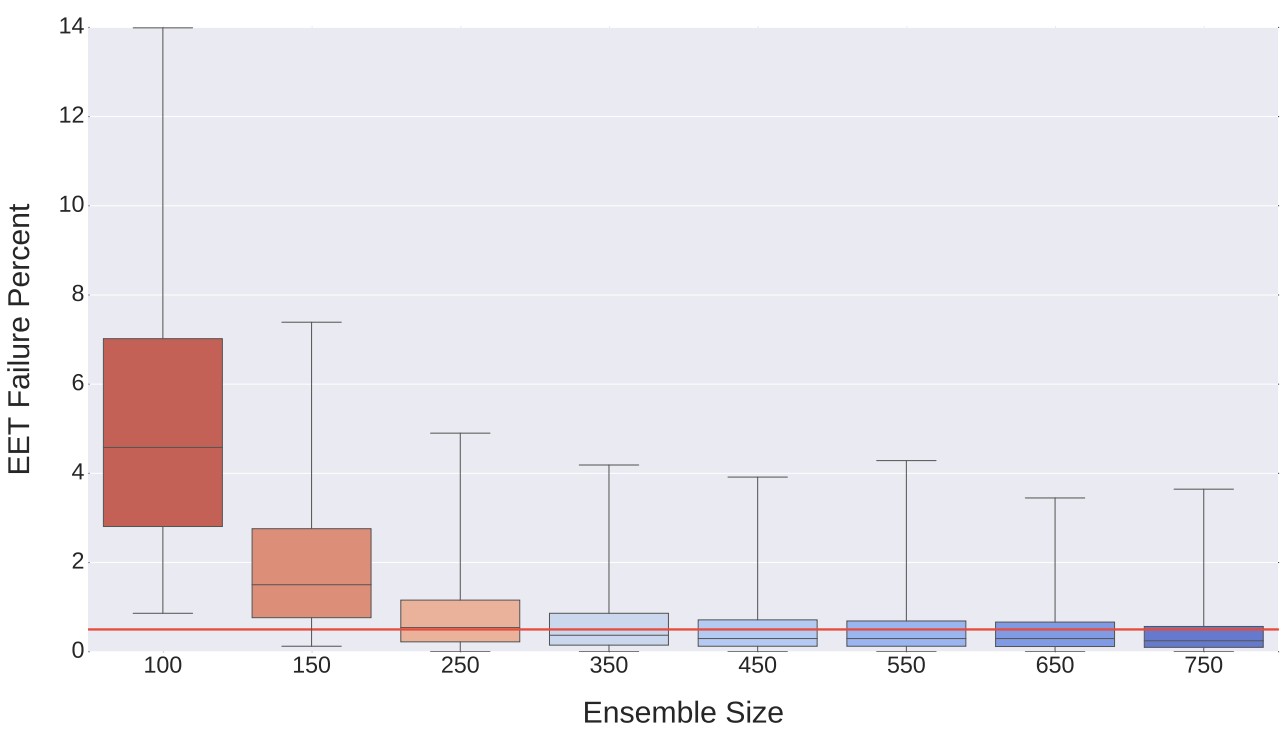

**Figure 3.** Box plot of EET failure rate distributions as a function of ensemble size. The distributions are generated by randomly selecting a number of simulations (ensemble size) from a set of 801 simulations to compute PC loadings. From the remaining set, 30 simulations are chosen at random. These simulations are projected into the PC space of the ensemble and evaluated via EET. For each ensemble size, 100 ensembles are created and 100 experimental sets are selected and evaluated. Thus each distribution contains 10,000 EET results (40,600,000 total tests per distribution).



**Figure 4.** Each box plot represents the statistical distribution of the difference between the global mean of each variable and the unperturbed, ensemble global mean, then scaled by the unperturbed, ensemble global mean for both the 30 ensemble members and 30 CLM_ALBICE_00 members. The plots on the left are generated from nine time step simulations, while those on the right are from one simulation year.