# Peer review of "Nine time steps: ultra-fast statistical consistency testing of the Community Earth System Model (pyCECT v3.0)"

_Geoscientific Model Development, 2017_

## Referee Comment (RC1) · Anonymous Referee #1 · 6 Jun 2017

The article by Milroy et al. is about a methodology to verify that a given Earth System Model produces (or not) statistically undistinguishable climates when, for instance, it is used on two different computers, or with different compilation options. In general, their methodology applies to cases when Bit For Bit verification is not possible because the rounding differences or treatment of the operations causes very small differences at the first time step that then grow due to the inherent chaotic nature of the model. Their approach is pretty unexpected in my point of view since I would personally rely on very long climate simulations to address the same problem. But their methodology is really convincing and the results they obtain prove that their approach is relevant. Each step of the methodology is both well described and robust, and I agree with the

interpretation of the results (smartly limited to the actual implications of the results and applicability of the methodology). Overall the scientific quality of the paper is really good and it is well written. With only one two minor comments, I highly recommend the publication of this article that perfectly fits within the scope of GMD and hope that pyCECT will be adapted and used in many climate modeling centers in the future.

Minor comments: Page 4 line 3-10: the description of the PCA decomposition in section 2.1 would deserve more detail (step by step as in Milroy et al. 2016) to be more quickly understandable.

page 4, line 24: replace 'studies' with 'study'

––––––––––––––––––––––––––––––––

---

## Referee Comment (RC2) · Anonymous Referee #2 · 12 Jun 2017

General Comments:

Software testing, which targets to detect and fix the bugs in the software as early as possible, is an essential step in software development, while the efforts for the software testing of Earth system models (ESMs) seems far from enough, especially when the code lines of ESMs expand rapidly and the climate simulation results still contain obvious uncertainties, which make model developers more difficult to successfully detect software bugs and then fix them. In another aspect, the computer environments (including compilers, processor architectures, etc.) always have to be upgraded with the fast advancement of technologies. Due to the chaotic nature of the climate system

as well as ESMs, bitwise identical outputs may not be obtained after porting an ESM to a new computer environment. Therefore, non-bit-for-bit (BFB) consistency testing are highly desirable for model development.

This paper proposes an ultra-fast consistency testing approach for CESM (pyCECTv3.0), based on the authors' previous works CAM-ECT (Baker et al., 2015; Milroy et al.). This new approach UF-CAM-ECT is easy to understand and is potential to significantly accelerate the software testing, and this paper is well motivated. It of course is prospective to improve the model development. Considering there are some points in the papers are not clear enough currently, I recommend a major revision before the further consideration for publication.

Specific comments:

1. As it is difficult to theoretically prove the effectiveness of UF-CAM-ECT, representative test cases are required for evaluation. I note that this paper follows some test cases used for the CAM-ECT (Baker et al., 2015; Milroy et al.), while many tests cases of porting validation that have been used for evaluating CAM-ECT are not used in this paper. Considering porting validation is an important usage case for UF-CAM-ECT, I suggest authors use more test cases of porting validation in this paper, such as most of the corresponding test cases used for CAM-ECT (especially the cases with change of processor architectures).

2. UF-CAM-ECT in this paper specifically uses the output after 9 time steps of model simulation for testing. The number of "9" is generally motivated from Figure 1 that shows the sensitivity of output fields to the number of time steps. I agree that Figure 1 is a good motivation for the number of "9", however, I am afraid that the authors' statement that "While nine time steps may not be strictly optimal, we have no reason to believe that more time steps results in a more accurate ensemble consistency determination" (P5L24) is not convincing enough, because Figure 1 cannot represent the testing results of UF-CAM-ECT, which means that this paper does not show that "9"

is a "right" number of time steps according to the testing results of UF-CAM-ECT. To further prove "9" is a "right" number or to find a "right" number, I suggest authors evaluate the consistency of the testing results between different numbers of time steps, for example, gradually increasing the number of time steps from a number smaller than 9 (6?) to a number larger than 9 (45?).

3. The cost of UF-CAM-ECT depends on the ensemble size. Given the number of "9" of time steps, the ensemble size is much larger than the ensemble size involved in CAM-ECT. It is unclear about the relationship between the number of time steps and the ensemble size. However, it may be guessed that a larger number of time steps may require a smaller ensemble size. If that is true, it is possible that the cost of UF-CAM-ECT may become smaller with a bigger number of time steps, especially when the cost of model run always is not linear with the number of time steps because the initialization cost of a model run is significant and generally is unscalable with the increment of processor cores. More evidences about this point are welcome.

4. It is understandable that UF-CAM-ECT may have different test results from CAM-ECT, for example, the test results shown in Table 3. However, it is still a challengeable situation that UF-CAM-ECT may issue a pass when CAM-ECT issues a fail, for example, the manufactured examples CPL_BUG and CLM_HYDRO_BASEFLOW in Table 3. I do not fully agree the authors' statement that "in practice the two examples we gave in Sect. 5.3.2 were quite contrived as we could not identify more realistic ones" (P13L5), because similar modifications may really happen in model development or research. For example, scientists may only change the land surface data of several grid points when simulating the atmosphere for some scientific researches, or changing a few ocean grid cells into land grid cells in coupled climate model simulations. So it may be risky to state that "Therefore in practice, applying CAM-ECT as a second step should only be considered when UF-CAM-ECT issues a fail" (P13L9) and that "The ultra-fast test is cheap and quick, and further testing is not required when a passing result indicating statistical consistency is issued" (P14L1). More evidences or discussions about how to make users safely trust the passes issued by UF-CAM-ECT without the loss of chances for bug detection based on CAM-ECT are welcome.

5. Considering UF-CAM-ECT is prospective to be general to CAM in various simulations and even general to various models, it will be interesting to know whether UF-CAM-ECT keeps the same results or even the "same" failure rates for the same set of test cases under different simulations with different input data, different parameterization schemes, different time steps, or different resolutions.

6. It will be welcome if authors list out the failure rates and the failure results at the same time in each table.

---

## Author Comment (AC1) · 19 Jul 2017

Thank you for the encouraging review and suggestions. The two comments are addressed below.

1. *Page 4 line 3-10: the description of the PCA decomposition in section 2.1 would deserve more detail (step by step as in Milroy et al. 2016) to be more quickly understandable.*

   We agree that this section benefits from a more thorough explanation. We have updated section 2.1 to include a modified version of the description from Milroy

et al. 2016.

2. *page 4, line 24: replace 'studies' with 'study'*

   Thank you; we have made this correction.

---

## Author Comment (AC2) · 20 Jul 2017

Thank you for the thorough review and helpful suggestions; we address each item below.

1. [. . .] *I suggest authors use more test cases of porting validation in this paper, such as most of the corresponding test cases used for CAM-ECT (especially the cases with change of processor architectures).*

   We appreciate the suggestion that porting examples may be of particular interest, and have included more porting experiments in the revision.

[Figure]

2. *I agree that Figure 1 is a good motivation for the number of "9", however, I am afraid that the authors' statement that "While nine time steps may not be strictly optimal, we have no reason to believe that more time steps results in a more accurate ensemble consistency determination" (P5L24) is not convincing enough, because Figure 1 cannot represent the testing results of UF-CAM-ECT, which means that this paper does not show that "9" is a "right" number of time steps according to the testing results of UF-CAM-ECT. To further prove "9" is a "right" number or to find a "right" number, I suggest authors evaluate the consistency of the testing results between different numbers of time steps, for example, gradually increasing the number of time steps from a number smaller than 9 (6?) to a number larger than 9 (45?).*

The major reason we present Figure 1 is to demonstrate sensitive dependence on initial conditions in CAM and to show that choosing a small number of time steps may provide sufficient variability to detect statistical difference resulting from significant changes. In fact, Figure 2 provides the stronger argument for choosing 9 time steps: we do not have any evidence to suggest that an ensemble formed from, e.g., time step 45 will contain variability that improves the UF-CAM-ECT sensitivity or classification accuracy. Since the behavior of most CAM variables (especially those resembling the top two in Figure 2) is consistent through time step 45, the plot indicates that there may be no advantage to using more steps. While some variables do manifest a similar trend to that of the bottom row (increasing variability with time step), integrating that greater variability into an ensemble does not necessarily translate to a more accurate test. In fact, choosing a smaller number of time steps is advantageous from the standpoint of capturing the state of test cases before CESM feedback mechanisms can take effect. We acknowledge that we did not sufficiently explain our choice of time step and have updated the text in section 2.2 in light of this

comment. We appreciate the reviewer highlighting this point.

3. *The cost of UF-CAM-ECT depends on the ensemble size. Given the number of "9" of time steps, the ensemble size is much larger than the ensemble size involved in CAM-ECT. It is unclear about the relationship between the number of time steps and the ensemble size. However, it may be guessed that a larger number of time steps may require a smaller ensemble size. If that is true, it is possible that the cost of UF-CAM-ECT may become smaller with a bigger number of time steps, especially when the cost of model run always is not linear with the number of time steps because the initialization cost of a model run is significant and generally is unscalable with the increment of processor cores. More evidences about this point are welcome.*

The relationship between model time step number and ensemble size is unlikely to be strictly inverse. For example, in Milroy et al. 2016 we concluded that ensembles of size 300 or 453 are necessary for accurate CAM-ECT (12 month) test results.

We agree that model initialization and I/O overhead contribute to run time nonlinearity in the number of time steps. In fact, the majority of the cost of a nine time step run is due to initialization and I/O, as evidenced by the average run time of $t_1$ (96 seconds), $t_9$ (110 seconds), and $t_{45}$ (118 seconds). (The average at each time step was computed from 10 runs of 900 MPI processes and two OpenMP threads on Yellowstone.) As we noted above, while nine time steps may not be strictly optimal, we do not have evidence that more time steps would result in greater accuracy in ensemble consistency determination, or that forming an ensemble from a later time step would reduce computational cost by reducing the ensemble size.

Moreover, our primary consideration in this study is to find the smallest CESM time step that permits UF-CAM-ECT to evaluate experimental output in maximal agreement with CAM-ECT, and to detect small-scale changes via instantaneous global mean values before model feedback. An ensemble created at the ninth time step has these desirable properties. Optimizing the cost of ensemble generation and test evaluation is not a main consideration of this study, as UF-CAM-ECT is already an improvement of a factor of 70 over CAM-ECT in this regard. Any incremental improvement in UF-CAM-ECT would be negligible in this context.

4. *It is understandable that UF-CAM-ECT may have different test results from CAM-ECT, for example, the test results shown in Table 3. However, it is still a challengeable situation that UF-CAM-ECT may issue a pass when CAM-ECT issues a fail, for example, the manufactured examples CPL_BUG and CLM_HYDRO_BASEFLOW in Table 3. I do not fully agree the authors' statement that "in practice the two examples we gave in Sect. 5.3.2 were quite contrived as we could not identify more realistic ones" (P13L5), because similar modifications may really happen in model development or research. For example, scientists may only change the land surface data of several grid points when simulating the atmosphere for some scientific researches, or changing a few ocean grid cells into land grid cells in coupled climate model simulations.*

We did not mean to imply that similar modifications would not be performed for research and development purposes. We reworked the sentence as follows: "While discovering examples where UF-CAM-ECT issues a pass and CAM-ECT issues a fail is conceptually straightforward (e.g. a seasonal or slow-propagating effect), in practice none of the realistic changes suggested by climate scientists and software engineers resulted in a discrepancy between CAM-ECT and
UF-CAM-ECT. We constructed the two examples presented in Sect. 5.3.2 accordingly, which went beyond changes described as realistic by climate scientists and software engineers."

[. . .] *it may be risky to state that "Therefore in practice, applying CAM-ECT as a second step should only be considered when UF-CAM-ECT issues a fail" (P13L9) and that "The ultra-fast test is cheap and quick, and further testing is not required when a passing result indicating statistical consistency is issued" (P14L1). More evidences or discussions about how to make users safely trust the passes issued by UF-CAM-ECT without the loss of chances for bug detection based on CAM-ECT are welcome.*

We performed numerous experiments attempting to find examples of cases where UF-CAM-ECT issues a Pass, but CAM-ECT issues a Fail. We enlisted the help of climate scientists and elicited the input of CESM software engineers to conceive of examples of such a split decision. The only cases we found were those reported in our paper. We will add emphasis in the manuscript that for important applications a researcher could consider using both tests, but otherwise UF-CAM-ECT appears sufficient.

5. *Considering UF-CAM-ECT is prospective to be general to CAM in various simulations and even general to various models, it will be interesting to know whether UF-CAM-ECT keeps the same results or even the "same" failure rates for the same set of test cases under different simulations with different input data, different parameterization schemes, different time steps, or different resolutions.*

We agree that studying the failure rates across different scenarios is of interest. We have already successfully used CAM-ECT with both the Finite Volume (FV)

[Figure]

as well as the default Spectral Element (SE) dynamical cores. We have also used it with fully coupled (active ocean) models. We have not explored multiple resolutions, but plan to do so and have added these suggestions in the section on future work.

6. *It will be welcome if authors list out the failure rates and the failure results at the same time in each table.*

   We appreciate this suggestion and have added failure rates for UF-CAM-ECT to tables 1 and 2.

   ───────────────────────────

---

## Referee Report (RR1)

Thanks for the modifications based on the first round of review comments.

In this round of view, I'd like to give some specific comments regarding these modifications, as follows:

1. Regarding the modifications for the first comment from the first round of review, some failure cases in pointing validation should be given.

2. Regarding the modifications for the fourth comment from the first round of review, I do not fully agree that "in practice none of the realistic changes suggested by climate scientists and software engineers resulted in a discrepancy between CAM-ECT and UF-CAM-ECT". It is true that the changes in model code or computing environment can introduce changes to most of grid cells in model cases, but my example that "scientists may only change the land surface data of several grid points when simulating the atmosphere for some scientific researches, or changing a few ocean grid cells into land grid cells in coupled climate model simulations" truly happens in our model development. I think the reason for such kind of discrepancy between CAM-ECT and UF-CAM-ECT is because UF-CAM-ECT uses globally averaged result while 9 steps are not enough to propagate local errors to the whole grid. So I guess that such kind of discrepancy can be solved when not only using the globally averaged results. If the authors agree, discussions or even update in UF-CAM-ECT should be made.

3. Regarding the modifications for the fifth comment from the first round of review, the authors should give related results in the paper, especially when these examples are already available.

---

## Referee Report (RR2)

**Comments on the authors' answers to the review of anonymous referee #2**

Referee #2 shed interesting light on various points of the paper. It definitely adds significant value to the paper. I'm satisfied with the answers of the authors.
I recommend the publication of the paper after the last minor corrections to the second round of comments by referee #2.

---

## Author Response (AR2)

**Author's Response, Minor Revision Iteration**

**Nine time steps: ultra-fast statistical consistency testing of the Community Earth System Model (pyCECT v3.0)**

**Referee 2**

We appreciate your feedback and address each item below. All page and line numbers refer to the diff document.

1. *Regarding the modifications for the first comment from the first round of review, some failure cases in pointing validation should be given.*

   Thank you for suggesting the addition of examples of failing porting experiments. We have included results from the CESM run on Summit and Cheyenne supercomputers with Fused Multiply-Add (FMA) instructions and xCORE-AVX2 optimizations (which enable FMA instructions) in Sect. 5.1 (page 8, lines 28-32, and page 9, lines 1-6). Both machines fail CAM-ECT and UF-CAM-ECT when these instructions and optimizations are activated. FMA instructions were discovered to cause CAM-ECT failure in Milroy et al. 2016.

2. *Regarding the modifications for the fourth comment from the first round of review, I do not fully agree that "in practice none of the realistic changes suggested by climate scientists and software engineers resulted in a discrepancy between CAM-ECT and UF-CAM-ECT". It is true that the changes in model code or computing environment can introduce changes to most of grid cells in model cases, but my example that "scientists may only change the land surface data of several grid points when simulating the atmosphere for some scientific researches, or changing a few ocean grid cells into land grid cells in coupled climate model simulations" truly happens in our model development.*

   We agree that "realistic" is subjective and have rewritten the passage as

follows (see page 13, lines 31 and 33, and page 14, lines 2-3):

While devising examples where UF-CAM-ECT issues a pass and CAM-ECT issues a fail is conceptually straightforward (e.g. a seasonal or slow-propagating effect), in practice none of the changes suggested by climate scientists and software engineers resulted in discrepancy between CAM-ECT and UF-CAM-ECT. Hence, we constructed the two examples presented in Sect. 5.3.2, using changes which were formulated specifically to be undetectable by UF-CAM-ECT, but flagged as statistically distinguishable by CAM-ECT.

*I think the reason for such kind of discrepancy between CAM-ECT and UF-CAM-ECT is because UF-CAM-ECT uses globally averaged result while 9 steps are not enough to propagate local errors to the whole grid. So I guess that such kind of discrepancy can be solved when not only using the globally averaged results. If the authors agree, discussions or even update in UF-CAM-ECT should be made.*

The massive change to the soil hydrology baseflow rate in the CLM_HYDRO_BASEFLOW experiment does not propagate to the land surface and atmosphere by time step 9. For this experiment, an examination of spatial variation of atmospheric variables is unlikely to reveal patterns caused by the change. However, in the case of the CLM_ALBICE_00 experiment, a modification with highly localized effects (0.36% of grid cells- predominantly in Antarctica) is detected in the atmospheric variables' global mean values by the $9^{th}$ time step. Taken together, these two experiments make it unclear how to employ spatial analysis in the context of initial time step consistency testing.

Since submitting this manuscript, we have performed further work on finding root causes of statistical inconsistency in the CESM (see "Quality assurance and error identification for the Community Earth System Model" by Baker et al. in *Proceedings of the First International Workshop on Software Correctness for HPC Applications)*. This work involves following a determination of statistical inconsistency back to contributing sections of code. Additional information such as spatially dependent variability could help refine this process. However, since both UF-CAM-ECT and CAM-ECT use Principal Component Analysis based on global averages, extending this framework to incorporate spatial information is non-trivial. In fact, such an enterprise is likely an entire study in itself, such as Baker et al., 2016, which created the POP-ECT based on spatial information. An investigation into root causes of statistical inconsistency, potentially including spatial analysis, is work in progress and may require an entirely different testing method.

3. *Regarding the modifications for the fifth comment from the first round of review, the authors should give related results in the paper, especially when these examples are already available.*

UF-CAM-ECT and CAM-ECT have been used with both Finite Volume (FV) and fully coupled (active ocean) models for port verification as part of the CESM software engineering workflow for pre-release versions of CESM 2.0. Such results are not published as we cannot publish results on CESM model versions that are not yet publicly available. We do not have FV and fully coupled results for the version in this manuscript.

**Note:**

We discovered a minor indexing error in the code used to generate Figure 1, and have updated the plot. The trend and the conclusions to be drawn from the plot have not changed.

[revised manuscript text omitted]